# A local community on a global collective intelligence platform: A case study of individual preferences and collective bias in ecological citizen science

Ofer Arazy[1]*, Keren Kaplan-Mintz[2], Dan Malkinson[3], Yiftach Nagar[1,4]

**1** Department of Information Systems, The University of Haifa, Haifa, Israel, **2** Department of Learning and Instructional Sciences, The University of Haifa, Haifa, Israel, **3** School of Environmental Sciences, The University of Haifa, Haifa, Israel, **4** School of Information Systems, Academic College of Tel Aviv-Jaffa, Tel Aviv-Yafo, Israel

* oarazy@is.haifa.ac.il

**Data Availability Statement:** All relevant data are within the manuscript and its Supporting information files.

## Abstract

The collective intelligence of crowds could potentially be harnessed to address global challenges, such as biodiversity loss and species' extinction. For wisdom to emerge from the crowd, certain conditions are required. Importantly, the crowd should be diverse and people's contributions should be independent of one another. Here we investigate a global citizen-science platform—iNaturalist—on which citizens report on wildlife observations, collectively producing maps of species' spatiotemporal distribution. The organization of global platforms such as iNaturalist around local projects compromises the assumption of diversity and independence, and thus raises concerns regarding the quality of such collectively-generated data. We spent four years closely immersing ourselves in a local community of citizen scientists who reported their wildlife sightings on iNaturalist. Our ethnographic study involved the use of questionnaires, interviews, and analysis of archival materials. Our analysis revealed observers' nuanced considerations as they chose where, when, and what type of species to monitor, and which observations to report. Following a thematic analysis of the data, we organized observers' preferences and constraints into four main categories: recordability, community value, personal preferences, and convenience. We show that while some individual partialities can "cancel each other out", others are commonly shared among members of the community, potentially biasing the aggregate database of observations. Our discussion draws attention to the way in which widely-shared individual preferences might manifest as spatial, temporal, and crucially, taxonomic biases in the collectively-created database. We offer avenues for continued research that will help better understand—and tackle—individual preferences, with the goal of attenuating collective bias in data, and facilitating the generation of reliable state-of-nature reports. Finally, we offer insights into the broader literature on biases in collective intelligence systems.

**Funding:** This research was supported in part by the Data Science Research Center (DSRC), University of Haifa, Grant #100009444. Co-authors DM and OA are the recipients of this funding award. The funders had no role in study design, data collection and analysis, decision to publish, or preparation of the manuscript.

**Competing interests:** The authors have declared that no competing interests exist.

## Introduction

Crowds and groups of individuals working in coordinated manners, can exhibit collective intelligence—an emergent capacity to collectively learn, generate knowledge and insights, solve problems, make predictions, and produce artifacts [1, 2]. This collective intelligence is often harnessed through various digital platforms and technologies, such as Wikipedia or iNaturalist (with over 150 million observations as of August 2023), where users contribute, rate, and curate content [3, 4]. In Wikipedia, for example, diverse groups of people collaboratively produce high-quality articles, accurately and comprehensively representing a wide variety of topics [5]. On iNaturalist, communities of nature lovers report on plant and animal observations, collectively generating maps of species' tempo-spatial distribution [6].

Global collective intelligence platforms often organize people through networks of local or national groups. For example, at a global level, Wikipedia is organized around languages, operating over 300 language-specific communities, where a language is often associated with specific geographical regions (or nations) and cultures; and within each such community, work is sometimes organized around topic-specific projects (i.e. WikiProjects; [7]). Similarly, iNaturalist is organized as a network of national "nodes" (over 20 national representatives) and projects (each focusing on a particular species or region) [8].

Individuals in these communities have preferences and dispositions that, when aggregated, could potentially yield biases in the collectively-generated artifact (note: to maintain clarity, in this paper we tried to reserve the use of the term "bias" to denote an uneven or disproportionate representation of a particular subject or variable within the collectively-generated artifact, whereas when to individual people, we use "preferences" or "dispositions" etc.). For instance, when like-minded people come together to create a Wiki article, the article may not provide a balanced representation of the topic at hand [9]. Likewise, if nature observers on iNaturalist have a preference for a particular species, (e.g., a rare species, or a "charismatic" species, such as large mammals or colorful insects [10]) they are likely to over-report observations of that species, and similarly under-report other species' observations (e.g., species that are very common, or considered "uninteresting," such as flies), yielding a bias in the aggregate map of species distribution [11, 12].

Theories of Collective Action and Collective Intelligence suggest that under certain conditions, namely when the group is diverse in terms of members' knowledge, experiences, and perspectives, and when individuals' inputs are largely independent of one another, concerns for participation bias [13] and social influence [14, 15] are largely alleviated, and group members' biases could offset and collectively cancel one another out, yielding an unbiased outcome [16–19]. Nevertheless, eliminating bias and achieving unbiased outcomes in collective intelligence systems is difficult [20, 21]. Notably, the literature on Wikipedia discusses racial, cultural, gender, and other biases [22–25]; researchers have shown, for example, that Wikipedia's coverage of Western culture, geography, and history is much more comprehensive than its coverage of other cultures [20, 26–28]. Likewise, bias exists in other collective intelligence systems.

We investigated one form of such collective intelligence: contributory science [12, 29], or more specifically citizen science (CS), which allows scientists to leverage participant-generated data while providing an opportunity for engaging with local community members [30, 31]. CS is becoming a powerful means for addressing complex scientific challenges [32–34] and scientific breakthroughs, such as the achievements made by contributors to *Foldit*, an online game in which the crowd predicts chemically-stable foldings of proteins [35]. Our focus here is on CS projects for ecological monitoring, where CS presents an alternative to traditional professional protocols [31, 34, 36–42]. As smartphones became a ubiquitous commodity, their

widespread availability and adoption equipped many citizens with high-resolution cameras and GPS capabilities, which enable them to record geolocated images and videos. The data they record are collected on digital citizen-science platforms (notable examples include eBird and iNaturalist), which have become the largest source of biodiversity data [43]. These data allow professional scientists to address ecological and evolutionary questions regarding both geographic and temporal patterns [12]. In summary, the increased scope of CS projects for ecological monitoring in recent years has provided a new and important means for large-scale data collection [34, 41], which in turn plays a pivotal role in the efforts to conserve, manage, and restore natural environments [44–46].

The ability to use citizen-generated data (in particular, from unstructured monitoring) hinges on the quality of these data. Data quality is often defined in terms of its "fitness for use" [47, 48], a definition that is commonly applied to geographic information [49, 50]. In order to assess the "fitness for use" of citizen-generated biodiversity data, we need to consider the possible uses of these data. Recent research has demonstrated that CS data (including data from sporadic monitoring) can be valuable for addressing a variety of research questions, related to issues such as species' behaviors and traits, a species' spatial distribution, and species' richness (a count of distinct species) in a particular region: For example, CS data have been used to document the presence and range of rare species and morphs [51, 52] and to examine organismal responses to climate change and human activity [53–55]; digital applications such as eBird and iNaturalist are used to understand species demographics and range expansion [56, 57]; and finally, the availability of this plethora of biodiversity data has facilitated the development of a cost-effective framework for managing wildlife habitats and populations [58–61]. Although the possible uses of data determine its "fitness" and quality, often not all uses are known at the time of data collection [62]. We, thus, adopt a "use-agnostic" perspective [63, 64] and our analysis attempts to provide a broad assessment of observer-based biases that transcends any particular ecological research question.

## Challenges associated with CS data

Despite the potential benefits introduced by CS data, recurrent spatial and temporal biases can sabotage or dilute their applicability [12]. Especially with unstructured CS data, sampling locations and objectives are generally not predefined, and participants autonomously choose to collect data on certain organisms in certain areas and times. Would-be participants consequently select areas and environments they perceive as safe or have access to [65]. Also, areas of environmental justice concern (e.g., poor air and water quality, and high toxicant levels) are frequently underrepresented [66]. In particular, thus far, the biases inherent to CS data have prevented scientists from using unstructured CS data for estimating species abundance, which is particularly relevant for conservation aims [11]. The risks to data quality, and specifically the concerns regarding potential biases in CS data, have called into question the ability to use such data for research and practical purposes (e.g., developing conservation policies and intervention programs). Whereas CS has recently been employed for addressing some of these purposes, to date, other uses (specifically, estimating species' abundance) still rely almost entirely on traditional systematic monitoring protocols, as scientists are cautious of the potential data-quality risks in CS data [11, 12]. In sum, notwithstanding the potential benefits of CS, ensuring that CS-collected data are fit for scientific use poses a key challenge. Hence, a potential solution for environmental scientists and for collective intelligence systems in general is to eliminate biases and thus ensure the quality of the collectively-produced artifact [5, 34, 41, 67–70]. The need to understand the quality of CS data, and specifically the potential biases in these data, served as the impetus for this research.

In the context of CS for ecological monitoring, prior research has attempted to deduce the community's overall bias towards a species, time, or location of observations from the community's aggregate pattern of reported observations [11, 34, 41]; however, such attempts are inherently limited, given that often there is no other existing measure representing the species' true tempo-spatial behavior at a given moment (i.e., ground truth). Consequently, without having any objectively obtained measure for comparison, relying on the aggregate pattern of community-members' reports in effect risks mixing the notion of nature's true state with the preferences and choices of the people involved in the CS project [12]. Consider, for instance, the case where the aggregate pattern of reported observations shows a yearly increase in the count of a particular species, say wolves. This trend in the reported data may reflect an actual increase in the wolf population, or alternatively, it may reflect the community's growing interest in wolves (e.g., due to recent attacks on livestock). Likewise, a decrease in the number of yearly reports on a species may reflect either an actual drop in that species' numbers or, alternatively, the community's reduced interest in that species (e.g., because the species is considered extremely common). Deducing peoples' biases from reported observations is nearly impossible, suggesting that alternative methods should be employed for studying community-members' biases.

In this study, our aim was to pursue such an alternative, by providing a rich contextual description of peoples' preferences and biases, revealing the nuances of their considerations regarding the content they choose to record and share with the collective intelligence system. First, we need to examine what is known to date regarding how CS has been used and how the issue of bias has been addressed.

## Related work

In this section we review prior studies on biases in CS. Whereas at the individual level, we focus on personal preferences or constraints that affect community-members' contribution-related decisions, at the aggregate level, we are interested in the way in which these choices translate into biases in the collectively-produced artifact.

**Citizen science for ecological monitoring.** Citizen science has been applied to ecological purposes such as estimating species dynamics, mapping species distributions, and studying climate change ecology [44, 71, 72]. The majority of CS projects for ecological monitoring are nonsystematic and unstructured, i.e., some guidelines are provided but not imposed, such that participants are free to report on any specimen from any species they observe without any spatiotemporal restrictions (i.e., monitoring is opportunistic) [73]. That is also the case with many of the projects on the iNaturalist platform. Such an opportunistic monitoring approach stands in stark contrast to traditional structured monitoring, where observers are required to adhere to formal sampling protocols, which define all aspects of sampling events, including location, duration, timing, target species, etc. [74, 75]. Although unstructured projects usually benefit from wide participation due to their data collection flexibility, they are more susceptible to observer-based biases [34, 41, 42, 70, 76–78]. As a result, to date, scientists have been wary of using unstructured CS data, despite the potential benefit of wide participation [11, 12].

In an effort to address this situation, researchers have started to pay greater attention to observer-based biases in CS ecological monitoring projects [11, 12, 34, 41, 79–81]. More broadly, this body of research is related to the study of biases in collective intelligence systems [24, 82, 83]. Recent studies have attempted to account for these biases using various statistical approaches [12, 84–87], but these models do not consider the complexity of human social variables that create biases in these datasets [12]. We maintain that a key impediment to the development of robust bias-correction methods is an insufficient understanding of observers'

attitudes, preferences, and decision considerations. Understanding observers' monitoring and reporting behaviors can shed light on observers' decision-making process, as well as on the manners in which, collectively, individuals' choices may amplify or attenuate biases, and hence is essential for developing statistical bias-correction methods.

**Observer-based biases in ecological monitoring, CS projects.** The data reported to CS biodiversity platforms, such as eBird and iNaturalist, can be driven by social and ecological factors, leading to biased data. Though empirical work has highlighted the biases in CS data, little work has articulated how biases arise in CS data. The literature on CS ecological monitoring distinguished between biases that are associated with species-inherent properties (e.g. size and pattern of species, which influences their detectability), and observer-based biases such as those linked to observers' expertise, preferences, and monitoring equipment [12]. Our focus here is on the latter: observer-based biases. Observers' considerations could be broadly classified into three categories: temporal, spatial, and species-related (or taxonomic) biases [76]. Thus, observers' reports may be spatially clustered due to ease of access to some areas, such as proximity to the observer's residence or commuter route [88–90], or difficulty accessing other areas [77, 91]. Such reporting patterns yield spatial redundancies or gaps in the collected data [79]. Similarly, observers' temporal activity patterns and their preference for certain species may introduce additional biases [34, 77, 78]: the fact that more people are active during the day can lead to gaps in reporting of nocturnal species. To date, the discussion of biases in the literature has been primarily conceptual, lacking an empirical investigation of observers' attitudes, preferences, and choices. An exception is Bowler et al. [92], who used a questionnaire to study citizen scientists' decision-making processes when recording species observations. They focused on factors related to observers' motivations, experience, and knowledge; however, they did not directly investigate preferences and biases.

In sum, although previous relevant research has acknowledged the importance of observer-based biases in CS ecological monitoring projects, there is a paucity of human-centered studies that investigate observers' specific considerations. In an attempt to address this gap in the literature, our study posed two primary research questions: RQ1–what are observers' considerations when deciding where, when, and what to observe, as well as which observations to report? And RQ2–to what extent are there commonalities in observers' considerations? We recall that the aggregate pattern of community-members' preferences and/or constraints has immediate implications for the consequential reliability of the database of reported observations, and—as a result—to scientists' ability to gain reliable, actionable insights from the data.

## Materials and methods

In order to address these research questions, we wanted to closely study the mindset of citizen scientists of a local community. We chose a particular citizen-science community, that allows its members extraordinary levels of autonomy, i.e., affording them to report on any species they choose, at any place or time, and only providing limited guidance and direction (corresponding to the project's goal of representing species' spatiotemporal distribution). We assumed that such a setting would likely expose a broad range of observer motivations, attitudes and preferences, and allow us also to reveal commonalities that could turn into collective biases in the data they report.

We conducted a multimethod case-study [93, 94], over the course of four years, collecting data from observations, questionnaires, interviews, and archival textual material. This prolonged, in-depth investigation of the project enabled us to provide a streamlined and comprehensive view of the community members' citizen-science practices, capturing observers'

perceptions and attitudes [94–96]. The data collected for this study was qualitative, and was analyzed using thematic analysis methods [97].

## Research setting

The setting for this study is "Tatzpiteva" (in Hebrew, a compound of "nature" and "observation"), a CS project that is unrestricted in its biological scope, allowing observer-based preferences to manifest. That is, the observation protocol is unsystematic and opportunistic, as opposed to systematic monitoring that is commonly used in scientific research, whereby observers are free to choose the species, time, and location of observation. Tatzpiteva, launched in January 2016, is a local citizen-science initiative which focuses on a rural area the size of 1,200 square km in Israel's northern region, where residents live in small communities (the only town in the area has a population of 7,000) and the dominant land use is open rangelands. The project is operated by the regional council together with the University of Haifa. Observations are reported by a local community of volunteers. A part-time employee of the regional council who is an expert naturalist, works as the community manager, encouraging participation, curating the volunteered observations, and educating observers on nature-monitoring procedures. In particular, the community manager encourages the reporting of all species, so as to provide a representation of the region's biodiversity.

Tatzpiteva employs the iNaturalist online CS platform (https://www.inaturalist.org/) [98], whereby observers use a mobile phone (both Android and iPhone applications) and a website. In addition, Tatzpiteva (https://www.inaturalist.org/projects/tatzpiteva) has developed its own localized mobile application and web site, and data is transmitted to the iNaturalist platform via an API. Observations are recorded using a camera and then reported (or uploaded) to the online database; when using a smartphone app, recording and reporting are performed at once (unless limited internet connection delays upload); and when using a standalone camera to record observations, reporting is performed at a later stage via the website. During the time of our study, roughly 40,000 observations were reported on Tatzpiteva by 400 observers, making up roughly half of all the iNaturalist observations in Israel. Most of Tatzpiteva's observations were contributed by the community's core members, whereas the majority of members are peripheral and contribute only occasionally. The Tatzpiteva community of citizen scientists is also very active in the physical sphere, with face-to-face gatherings (e.g., biannual community meetings, exhibitions of observers' photos) and nature-observation field trips (e.g., on topics such as mushrooms or animals' tracks), which are organized by the project's staff, as well as by volunteers.

## Participants

The composition of online communities is often described in terms of core and peripheral members. While there is no single accepted definition of a community's core, the literature discusses core members in terms of their activity pattern (commonly, a small group of core members is responsible for the majority of the work), their tenure within the community, and by the roles and responsibilities they take [99] (for example, becoming a "curator" on iNaturalist).

During our study, we studied the community-at-large by participatory observations in many meetings over the course of four years, as well as by analyzing archival materials. But the bulk of this research was focused on the community's core members. We noted that there were 38 members who constituted the core of the community: these were highly-active observers (with a minimum of twenty-five observations reported), held special responsibilities and, at the time of our study, had been active for at least six months. These members were all sent a

questionnaire (described hereafter), and 27 of them signed an informed consent to participate in the study, and answered the questionnaire.

Tracking the online profile of those 27 participants on the iNaturalist platform revealed that they were responsible for 82% of the recorded observations in the entire Tatzpiteva project. Eight of them had been formally assigned "curator privileges" in the Tatzpiteva project, a position that corresponds to an administrator status in other online communities.

In the next stage, 15 of the questionnaire responders were interviewed, six of whom held curator responsibilities.

## Data sources

Our acquaintance with the Tatzpiteva project began at its inception in 2016. Over the course of four years, we spent an average of two weekly hours informally both viewing co-located activities (meetings with the project administrators and community meetings) and reviewing online activities (reports on the Tatzpiteva website). We thus accumulated about 400 hours of informal observations. These immersive experiences allowed us to gain a deep and intimate familiarity with the Tatzpiteva project and community, and to accumulate substantial formal and tacit knowledge regarding its procedures, governance, and community aspects. The rich knowledge gained from these sessions provided the context for understanding and interpreting the qualitative data that we later collected in more formal, systematic manners.

In general, we used archival textual materials as a source for background information about the community and how it works and functions, and both the questionnaires and the interviews provided the data pertaining directly to the research questions posed.

**Archival materials.** Archival textual materials were used as a source for obtaining information about the community and its workings, and specifically about the process by which the project leaders planned and guided the community's activity. These materials included the original funding proposal (November 2014), three yearly reports by the project's ecologist, and periodic newsletters issued by the community manager.

**Questionnaires.** Data for addressing our research questions, in particular, data regarding observers' considerations as to where, when, and what to observe, as well as which observations to report, were based on questionnaires and on follow-up interviews with focal community members. The questionnaire focused primarily on observers' species-related dispositions. It included closed questions regarding observers' demographics, activity frequency, and preferences, as well as three open-ended questions regarding participants' criteria for selecting what species to record. In particular, two of the questionnaire's open-ended questions asked participants: (Q3) How do you decide which observations to report and which to omit? What are the criteria you consider? (Q4) Are your observations oriented towards a particular species? If yes, which one? Do you actively go out into nature in an attempt to detect and record these species?

In addition, we introduced several questions regarding observers' reporting behavior. In order to ground these questions, we asked participants to focus on a limited number of species that are common in the region and thus participants had likely encountered them. To sample a broad range of preferences, constraints, and behavior, we selected species that vary in terms of their *detectability* (the observer's ability to notice the animal when it is nearby; determined by such factors as animal size, skin pattern and camouflage, as well as its general vigilance and its fear of humans or lack thereof), *recordability* (observers' ability to record an animal once detected; this is affected by a number of factors, including the animal's speed, as well as photography equipment), and *rarity*. Thus we opted to anchor the questionnaire, by focusing on four species—tortoise (*Testudo graeca*), wild boar (*Sus scrofa*), mountain gazelle (*Gazella*

*gazelle*), and golden jackal (*Canis aureus*), which vary along the aforementioned dimensions of detectability, recordability, and rarity. For each of these four species, the third open-ended question asked: (Q5): During your monitoring activities, how likely were you to have observed [this species] yet refrained from reporting this observation? What was the reason for not reporting this observation? This was preceded by a close-ended question, (Q2): Please rank the animals observed by Tatzpiteva participants to reflect your preferences or the strength of your or emotional connection to each (1 = most preferred; 9 = least preferred, regarding nine species that are common in the region: *jackal*, *wild boar*, *tortoise*, *porcupine*, *mole rat*, *hedgehog*, *fox*, *gazelle*, *and mongoose*). The list of questionnaire questions is included in S1 Appendix.

**Interviews.**   The questionnaire was followed-up with interviews. A member of the research team conducted and recorded the semistructured telephone interviews that were held with 15 highly involved community members (six of the interviewees had curator privileges). Interviewees (all of whom had completed the abovementioned questionnaire) were selected based on the community leader's referral. The interview lasted approximately 15–20 minutes. The goal of the interviews was to shed light on the broader context of participation, and expose preferences that are related to the locations and times of observations. The interviews also probed participants on their patterns of activity and views regarding Tatzpiteva's unstructured monitoring protocol. The guideline for semistructured interviews is included in S2 Appendix.

To summarize our investigation, after embedding ourselves in the Tatzpiteva community, we proceeded to collect data from the questionnaires and interviews. The relevant data set included all the data from the questionnaires and from the sections of the interviews pertaining to the constraints and preferences that shaped the participants' decisions regarding when, where, and what to observe and how they decided what they included in their reports. As all of the collected data were in Hebrew, a translation into English was provided by two members of the research team, both bilingual native-level speakers of English and Hebrew).

## Data analysis

In analyzing the questionnaire data, we followed the thematic analysis method [97, 100–102]. In accordance with the goals of our study, we focused on a detailed description of the particular qualitative themes that reflect the participants' thoughts regarding their criteria for choosing where, when, and what to observe, as well as which observations to document and report.

We performed the thematic analysis in two steps, beginning with the questionnaires, and then continuing to interviews. In the absence of a solid theoretical framework regarding the factors that influence observers' reporting decisions, which could have guided a theory-driven investigation, we performed an inductive, bottom-up thematic analysis, such that themes were directly derived from the data. Given the inductive nature of our thematic analysis, we were careful not to delve deep into the literature at this stage, so as not to approach the data analysis with preconceptions. The thematic analysis was performed independently by two members of the research team. Upon completion of their analysis, their resulting thematic maps were compared and discussed. We found the independent analyses to be highly consistent, wherein the key difference—beyond the wording of codes—centered on whether to consolidate two closely-related codes into a single composite code. In addition, there were four cases in which the meaning of the text was not entirely clear, which led to disagreements regarding the code that best corresponded to each of these text segments. The researchers discussed these inconsistencies until a consensus was reached. The result of this process was a set of agreed-upon codes and their definitions, the association of text segments to codes, and a grouping of codes to higher-order themes.

Next, we transcribed the interviews and applied the themes that had emerged earlier to our analysis of interview contents. The analysis of interviews, too, was initially performed independently by two members of the research team, and then consolidated through discussions between the researchers. The codebook is included in S3 Appendix.

Finally, through discussions, and relying on our intimate, unmediated acquaintance with the community, we synthesized findings and insights from all sources (questionnaires, interviews, and archival data) into emerging themes which we present next.

## Results

The presentation of results in this chapter is organized according to the emerging themes we identified through the process described above. As common and recommended in qualitative research, and specifically in ethnographic studies, we illustrated our findings with archetypal quotes that serve to highlight common attitudes and behaviors of the community members [103–106]. Quoted members are designated with brackets (e.g. [pr5] indicates participant #5).

### Background and descriptive statistics

Analysis of archival data, as well as interviews with the community members and administrators, shed light on the ways in which the project administration sought to shepherd the activity of the local community of observers. Whereas the nature-monitoring protocol that was used was entirely opportunistic, the project's administrators attempted to channel observers' participation by encouraging them to record all species, everywhere, at any time. Observers were encouraged to put aside personal preferences or any assumptions as to what is important (e.g., rare species) and try to record any species, across the entire area during all seasons and times. To wit, the community manager is quoted in the regional newspaper saying:

*At the highest level, the public decides what to monitor, performs monitoring, and takes part in drawing conclusions. There is no requirement to photograph only rare species, but [rather] also what seems trivial: crows and sparrows, wild boar and porcupines, wolves and chrysanths, oak and pine trees. This way, we will learn to know the entire [ecological] system . . ..*

*[May 2016, the regional council' newspaper].*

A year later, the community manager was interviewed for the same newspaper and added:

*Observers ask me: what should I record? and I respond: in order to deeply study the region's nature, we should record everything!—From ants to vultures, flowers, and every species in nature—they are all part of the ecological system . . . Furthermore, when we are taking a walk in nature, we see a plant and a few strides later we see the same plant again—should one upload another observation? The answer is Yes. Beyond species richness, we are also studying species abundance. . . .*

*[July 2017, the regional council' newspaper].*

The results regarding observers' considerations are based on the responses of the 27 participants who returned the questionnaires; a statistical summary of their characteristics is shown in Table 1.

**Table 1. Descriptive statistics of questionnaire participants.**

| Characteristic | Range | Mean | Median | St. Dev |
|---|---|---|---|---|
| Age | 34–77 | 50 | 45 | 11.73 |
| Number of reported observations | 27–3,514 | 558 | 131 | 839.2 |
| Gender | 16 men and 11 women | | | |
| Tools used | 15 members used a smartphone (13 used smartphones running the Android operating system, and 2 members used iPhones). | | | |
| | 6 members exclusively used only a standalone camera, uploading photos through the website. | | | |
| | 6 members switched between the use of a smartphone and standalone camera. | | | |
| Applications used (smartphone or website) | 2 members used iNaturalist's standard application. | | | |
| | 25 members used Tatzpiteva localized application. | | | |

### The primary factors underlying participants' reports

Altogether, we thematically analyzed 211 utterances that were extracted from questionnaires, and grouped them initially into 20 subcategories and then later into four themes, namely, the primary factors underlying participants' decisions of where, when, and what to observe, as well as which observations to report: (a) *recordability*, i.e. the ability to record the species once observed; (b) *community value*; (c) *personal preferences*; and (d) *convenience*.

**Recordability.** Many of the questionnaire participants pointed to constraints in their ability to record species that have been detected—i.e., *recordability*—as a key factor. Constraints were linked to two primary factors: First, certain species are easier to record than others, namely species that move slower and are less sensitive to the presence of humans, such as plants or slow-moving animals, e.g., *tortoise* (in contrast to shy and fast-moving animals, such as *gazelles* or *jackals*). A typical statement made by observers when describing the factors influencing their reporting process is: "The ability to take a picture" [pr3]. An example of a response explaining why a participant did not report on a particular species: "Animals that run away faster than it is possible to photograph [are] not reportable . . ." [pr4]. Second, *recordability* was heavily influenced by the observers' photography equipment and its ability to capture the distinctive features that facilitate accurate identification of the species, i.e., those using professional cameras were able to photograph at a distance. Typical statements describing observers' reasons for not reporting a detected species include: "Difficult to photograph using a smartphone" [pr7] and "inappropriate [camera] lens" [pr8]. Overall, in the context of our questionnaire, *recordability* emerged as the primary factor affecting an observer's recording behavior, with 76 utterances (36% of 211 utterances).

**Value to the community.** *Community value*, that is, the extent to which reporting the species is considered valuable for other members of the community or to the project's goal of creating an archive of the region's biodiversity, emerged as a key factor that influenced the decision of where, when, and what to observe, as well as which observations to report. *Community value* was associated with either the ability to accurately identify the species in photos; species' rarity or abundance in the region; the importance the observer ascribed to the archive of observations, or other assorted importance-related considerations. First, we found that the ability to identify the species was a key consideration in deciding what to report. Some participants mentioned their own ability to identify (or name) the species, whereas others indicated that they take into account the community's ability to identify the species. For example, [pr10] wrote: "I report everything that I see and I know it would be possible to identify. If the photo was of low quality or it was not possible to identify the observation (for example, a plant with no flower or fruit) I gave up. I have learned to differentiate between the identifiable features and those that are not identifiable" [pr10].

Second, we found that the extent to which observations are rare was a primary reason for reporting on a detected species, and conversely species' abundance was a key reason for opting not to record or report an observation. Survey participants frequently mentioned "rare" or "rarity" explicitly in their considerations for recording observations. Along the same lines, participants indicated that they were less likely to record "Species that are highly abundant—a crested lark is an abundant species and nobody reports it. Same as sparrows. The same goes for wild boars and gazelles" [pr1].

Third, a few participants mentioned that they chose to record an observation when it was important for other members of the local community or to other viewers of the data. Typical justifications for recording an observation included "An observation that I think may interest others . . ." [pr3] and "[the] importance of the information to the general knowledge base" [pr16]. In the interviews, three participants pointed to specific collective concerns related to the future development and the potential danger to nature, specifically the deployment of wind turbines in the region, as a reason for monitoring that particular area. For example, [pr10] stated the following:

> It may be possible to link [Tatzpiteva monitoring] to an environmental and social issue. For instance, now there is this issue of wind turbines in the region, we could conduct a [monitoring] project focusing on this issue, monitoring biodiversity in the area, which could serve as the basis for discussion and decision-making. That would be valuable from both the environmental and the communal perspectives.

> [pr10]

Lastly, questionnaire participants brought up additional *importance*-related considerations. Notably, several highly-active observers exhibited an understanding of systematic monitoring protocols (although not formally required in projects such as Tatzpiteva) and mentioned that their reporting decisions did not privilege any specific species, such that they recorded everything they encountered, for example, "I report everything, but mostly mammals and birds" [pr17]. Overall, in the context of our questionnaire, *community value* was found to be a key factor affecting observers' recording decisions, with 45 utterances (22% of 211 utterances). Rarity was the primary factor in this category, providing roughly half of all utterances associated with *community value*.

**Personal preferences.** Observers' *personal preferences* also had a substantial effect on participants' reporting decisions. We identified several categories for these preferences: personal preference for a particular species, region, or time; attraction to a species' specific features (e.g., the smell of a particular flower); the desire to learn more through others' feedback; limiting one's recurrent reporting of a particular species; and other personal preferences. First, many observers simply indicated that they chose to report what they personally found interesting and they avoided reporting what they found to be uninteresting. Related to this, a few participants mentioned emotional attachment or a personal previous experience (e.g., ". . . mostly, personal experience . . ." [pr8]) had influenced their reporting decisions.

Second, several participants mentioned that they chose to report species that they found attractive and beautiful (e.g., have a preference for what is "beautiful. . . awesome . . ." [pr3]) or have special features (e.g. "plants—color, size . . . will cause me to take more photos" [pr17]).

A third type of a *personal preferences* was the desire to learn, and a few questionnaire participants mentioned that they chose to record ". . .Things that I do not know and want to know . . ." and "I upload [images of] plants that I'm interested in knowing their name . . ." [pr14].

A fourth personal preference was the desire to avoid reporting the same species multiple times (e.g., [a consideration is] "The number of times that I reported the species in the past" [pr16]; [reason for not reporting] "tortoise—too trivial" [pr9]) Finally, some participants indicated their preference to report certain species or places, without providing an explanation. Overall, in the context of our questionnaire, 62 utterances (29% of 211 utterances) indicated *personal preferences* as influencing observers' decisions regarding what to record.

**Convenience.** Another important factor influencing observers' reporting decisions was the extent to which they found it convenient, in terms of the time and effort required to report an observation. For example, participants indicated that "I don't have time to engage with this" [pr2] and "[I don't have] spare time" [pr13]. *Convenience* is linked to several circumstances. First, some observers only make sporadic reports, such that observations are made when the person is engaged in a different activity—leisurely outdoor activity, professional work (e.g., a tour guide) or when driving—and is less attentive to reporting observations. For example, when describing her considerations regarding what to report, one participant mentioned "I report on observations while driving as part of my work . . . sometimes when I had some spare time on my way to work, I stopped to take a picture. I don't take a drive especially to make observations" [pr19]. Others mentioned that they do not carry a camera when in nature.

A second *convenience*-related consideration pertains to the equipment used, especially for observers who use professional cameras, and need to later upload those to the *Tatzpiteva* website, sometimes requiring them to edit and resize images, as evident in the following quote from an observer who uses a professional camera: "Lots of images taken, but only few are reported, for many reasons: lack of time to resize, sort, and uploading images [to the website]" [pr8].

Finally, *convenience* was also associated with location-based and temporal preferences, for example, observers indicated that they preferred to be active in certain places (e.g., not far from home; places that are easily accessible, in proximity to roads) and times (i.e., certain times of the day or week). For example, some indicated that they chose to monitor places with easy access or in a location with an abundance of species, "When I'm at a point with rich fauna, I'll document a large part of that fauna" [pr17]. Overall, in the context of our questionnaire, *convenience* was found to be an important factor affecting (28 of 211 utterances, constituting 13%) of observers' recording decisions.

## Summary of findings regarding observers' considerations

In sum, findings from our study provided in-depth insights regarding observer's considerations when deciding where, when, and what to report. Fig 1 below provides a summary of the count of utterances per each of the four primary factors discussed above. *Recordability* and *personal considerations* each made up roughly a third of the utterances regarding observers' decision of what to report (36% and 29% of the total 211 utterances, respectively), *community value* accounted for 22% of the utterances, and *convenience* had a lesser impact (13%) on observers' decision whether to report a species that had been detected. We note that participants often pointed to several factors that influence their decisions, and the average participant provided quotes that were linked to 2.5 of the four themes. The interviews provided a similar picture, with 14, 10, 12, and 7 interviewees addressing the themes of *recordability*, *community value*, *personal preferences*, *and convenience* (respectively).

Overall, we conclude that observers' decisions are affected by multiple factors, rather than by a single consideration. An illustrative quote from the questionnaire's response regarding

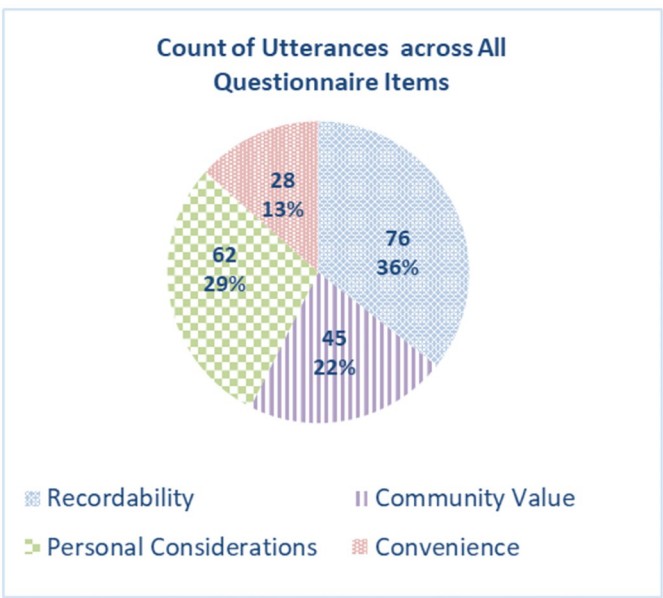

**Fig 1. A count of utterances for each of the factors affecting observers' reporting decisions.**

one responder's considerations: "Free time, rarity, the ability to take a high-quality photo" [pr13].

## Amassing individuals' preferences—Biases in the aggregate database

To understand the extent to which individuals' preferences and constraints amass to biases in the aggregate database of observation, we performed a secondary analysis on the labelled data. Going back to our original organization of utterances into subcategories and primary themes, we analyzed each subcategory and considered: (a) the extent to which each factor that influenced the decision whether to record or report an observation was common among the participants; and (b) whether such a shared preference/constraint could skew the aggregate database spatially, temporally, or taxonomically. Whereas answering the former was based directly on the labelled questionnaire data, determining the latter—i.e., the potential for aggregate biases—was also guided by the literature, the participants' behavior, and by the subject-matter expert (ecologist) in our research team. For example, when analyzing the subcategory labelled as "Photographing equipment enabling or inhibiting the ability to record" (under the theme of *recordability*), we found (a) that about two-thirds of the study's participants used a similar recording device, a smartphone; and (b) we determined that such a common pattern *was likely* to cause a taxonomic bias in the aggregate database (e.g., fewer photos of a species that can be seen only from afar), but *unlikely* to yield spatial and temporal biases.

Next, we summarized the data and identified the key risks to taxonomic, spatial, and temporal biases. In addition, we analyzed participants' ranking of their affinity to nine species (see Question 2 of the Questionnaire, in S1 Appendix), looking for common patterns. We present our findings below. As the next section demonstrates, in all three dimensions—spatial, temporal, and species-related (i.e., taxonomic), preferences and constraints were common—at least to some extent—among many of the participants, suggesting that the aggregate database of observations in the collective intelligence system is likely to be biased in particular directions.

**Temporal patterns.** Our analysis reveals several preferences/constraints that are commonly shared between community members, which may result in temporal biases in the

aggregate database. Specifically, almost all observations are reported during hours in the day where it is convenient for people to be out in nature: during daytime, especially in hours when the weather permits (not too cold; not too hot) and during weekends and holidays, where people have more time of leisure. This effect may interact with other factors, for instance, recording observations at specific times may require special photography equipment, hence the (lack of) availability of such special equipment may affect temporal activity patterns. For example, addressing the reasons for not reporting an observation, one member explained that "Encounters [with wild boar] are often in the dark or by surprise, and by the time the camera gets into action it's too late" [pr9].

Such a consideration is applicable to the majority of community members, who rely on smartphones for taking pictures. Additionally, our data indicate that participants often reported on observations only when they were able to identify the species with confidence. Here, again, there is an imbalance, with more reported data collected during the daytime, given that it may be more difficult to identify species in a picture that was taken during nighttime.

**Spatial patterns.** Our analysis identified a few preferences/constraints that were commonly shared among community members, which may result in spatial biases in the aggregate database.

First, *Accessibility* is a major issue. In the region that is at the focus of this study, accessibility is primarily affected by the following three factors:

1. *Place of residence*. People tend to report close to home, and more broadly—in more accessible places [107]. Hence, the uptake of contributory science platforms by observers is uneven across space and demography [12]. That is, when a significant portion of the community is concentrated in a particular city or village, we can expect a high volume of reports from that area.

2. *Terrain*. Some areas are hard to access on foot—these could be steep slopes, canyons, areas of dense thorny bush, etc. While these terrains are also less accessible to some animal species, there are plants, as well as some animal species, e.g., the *Rock Hyrax (Procavia capensis)*, which do inhabit such areas.

3. *Enclosed areas*. Some areas are enclosed by government or by private landowners, limiting all observers' access and thus their ability to report observations (we note that beyond the effect on observers' patterns of reporting, enclosure can also affect animals' spatial behavior patterns).

While accessibility was not often explicitly mentioned in response to the questionnaire, it was mentioned in some of the interviews. We suspect this may be because members did not think it was worth mentioning in writing, it did not occur to them, or it may have seemed obvious to them. However, the effect of accessibility on the breadth of the data is clearly evident in the participants' reports (please refer to [11] for a discussion on this matter). Second, when people share a concern regarding an ecologically harmful activity that takes place in a particular location (e.g., industrial development), they are more likely to monitor that area.

Lastly, tree and plant coverage may affect community-members' ability to identify the species in the image and, consequently, it might also reduce their tendency to report from these regions, where the vegetation allows animals to camouflage.

**Species-related patterns.** Individuals' preference towards particular species is a factor that greatly affects their reporting patterns. To the extent that community members share such preferences, the aggregate database may be crucially biased in terms of over- and under-representation of certain species. Some factors that are shared among community members exert a

**Table 2. Participants' affinity towards the various species, based on ranking averages.**

| Overall rank | Average rankings | Species |
|---|---|---|
| 1st | 2.8 | Gazelle |
| 2nd | 4.4 | Porcupine |
| 3rd | 4.5 | Fox |
| 4th | 4.7 | Hedgehog |
| 5th | 5.4 | Wild Boar |
| 6th | 5.6 | Tortoise |
| 7th | 6.1 | Mole Rat |
| 8th | 6.3 | Mongoose |
| 9th | 6.4 | Jackal |

A higher ranking on the list (i.e., a low numerical value) represents a stronger affinity.

particularly strong influence on observers' choices regarding what to monitor, posing a serious threat to the database's reliability.

*Affinity*. We first notice that people tend to feel some affinity towards certain species. As part of our questionnaire, we asked people to rank their personal affinity towards nine species. The results reveal common attitudes among community members, indicating a communal preference. Particularly, *Gazelles*, which were first on the list for nearly half of the participants, ranked first overall. In contrast, *jackals* were ranked last, with half of the participants ranking them in the last two spots. Table 2 below presents the order of participants' affinity for the various species, based on the average rankings.

*Attractiveness*. We noted a bias towards species that have a unique or attractive appearance, e.g., some butterflies, birds, and colorful plants. The large portion of observers that reported on these attractive species resulted in over-representation of these species in the aggregate database (and an under-representation of species with a more casual appearance). A common answer to the question regarding reporting criteria was: "Usually I report something that is unique" [pr6]. (Similar responses were provided by [pr1, pr2, pr4, pr11]). We suspect *Attractiveness* is one (probably a major) driver of *affinity* towards certain species.

*Rarity*. Our data show that preference for monitoring (looking for, and reporting) rare species was common among many members of the community. Species' abundance in the area is perhaps the most salient factor influencing observers' choices of what species to report: people are much *less* likely to report on common species. For instance, when answering "How do you choose what to report?" and "What factors are considered?" (Q1), [pr7] echoed the responses of many participants and answered: *"I choose to report rare species."*

Rarity is a species-related feature, so it is likely to affect all observers in a similar manner, and thus yield a major bias in the aggregate database, with an over-representation of the rare species. For example, the Tatzpiteva database includes more reports of porcupines than of ants (which are more abundant by several orders of magnitude [108]).

*Ascribed-importance*. Another related factor is the importance that the community assigns to specific species (e.g., gazelle is a national emblem; in northern Israel, wolves present a danger to livestock): the more community members share the beliefs regarding what is "important," the more skewed the aggregate database will be. For example, reflecting the attitudes of several community members, [pr9] stated: "[I report on] plants that people would encounter and would want to know their names."

*Species identification*. Participants noted that they often report on observations only when they are able to identify the species with confidence. A repeated theme was: "[I don't report

observations that] are very difficult to identify" [pr14]. On aggregate, this may lead to an over-representation of easily-identifiable species (e.g., mammals) and an under-representation of species that are more difficult to identify (e.g., mushrooms).

Finally, it is worthwhile pointing out another factor that might contribute to cumulative biases in the database, even though it is not directly related to people's affinities or attitudes towards wildlife, and that is the equipment they use.

*Equipment-effect*. Observers' photography equipment greatly influences the species that they choose to monitor and report. Professional equipment allows taking photos from afar, which is particularly important for recording observations of animals that shy from humans, as well as for small-size animals (e.g., insects). On the other hand, in some cases, smartphone cameras could be operated more quickly, allowing observers to capture images of animals that they come upon unexpectedly. To the extent that the majority of the community uses a single type of equipment (e.g., in our case, over two thirds of the participants used smartphone cameras), the aggregate database may be skewed, by including mostly animals that lend themselves to be recorded using that type of equipment.

It is worth noting that some of the factors discussed here are not constant, but rather may change over time. For example, as an observer amasses more observations of a particular species, he or she is less likely to report on this type of observation over time. Along the same lines, the importance that is assigned to species may shift over time, which suggests an interaction between species-related and temporal biases.

## Discussion

The underlying rationale for this study was the recognition that while unstructured CS projects better succeed in engaging the public and thus have the advantage of potentially producing large amounts of biodiversity data, they are also more likely to be vulnerable to bias. When each observer decides when, where, and what to monitor, as well as which observations to report, then to the extent that observers share preferences, views, values, etc.—which is especially likely in smaller, local communities—these individual considerations might accumulate to create collective biases, yielding archives of citizens' reports which do not reflect the actual spatiotemporal distributions of species in the environment [11, 12, 41, 42, 76–78, 89, 90, 109, 110].

Prior studies have raised concerns regarding the existence of such bias, and a few have even provided empirical evidence demonstrating that the data in the co-produced archives of observations were skewed [89, 90, 109, 110]. Yet, until recently, there has been a lacuna in academic literature on citizen science regarding the specific considerations and/or constraints that underlie observers' reporting decisions. A primary contribution of this paper is in bringing observers' authentic voices regarding their consideration as to where, when, and what to observe, as well as which observations to report. The findings from our case study surface the intricacies and nuances in observers' decision-making process.

In recent years, studies have introduced conceptual frameworks that describe citizens' monitoring process [11, 12], greatly contributing to our understanding of observer-based biases. These frameworks describe the monitoring and reporting process as a series of stages. Notably, Arazy and Malkinson [11] describe observers' decision-making process as: monitoring, identifying, recording, and reporting observations; and Carlen et al. [12] depict the process as a series of "filters", where each filter places a restriction on the possibility of reporting observed species, gradually moving away from the true state of nature (i.e. ground truth). The classification of biases that emerged from our empirical analysis largely corresponds to the stages in these conceptual frameworks. Specifically, our *Recordability* and *Convenience* categories

correspond to a large extent to Carlen et al.'s [12] *Detectability* and *Sampling* filters, respectively. Nonetheless, our findings challenge other aspects of these conceptualizations. One noticeable divergence is that the observer's decision-making process that emerged from our analysis is not a linear one; rather, observers' decisions intricately combine multiple factors and considerations, as illustrated by the following quote from an interview: "I report on an interesting observation. If there is an observation that seems to me as valuable. If there is something that I don't recognize, usually I report it. If these are things that are abundant, then it depends whether I feel like reporting it or not" [pr10].

Crucially, to the best of our knowledge, no prior study has linked individual observer considerations to biases in the aggregate database, or studied the conditions under which individual observer preferences cancel-out one another, which could provide a way to keep the aggregate database unbiased, or at least, less biased.

Our multimethod case study of a local community of nature observers—Tatzpiteva—on the iNaturalist platform sought to address this gap. The results pertaining to RQ 1 revealed the main factors that shape observers' reporting decisions, namely, *recordability*, *community value*, *personal preferences*, and *convenience;* identifying these categories enabled us to highlight commonalities in observers' considerations. The results of our analysis pertaining to RQ2 indicated that there is a real risk that some considerations are widely shared, yielding biases in the aggregate database of observations.

These findings not only inform and enrich the existing knowledge in the fields of CS and collective intelligence; importantly, they also suggest practical implications for both CS community custodians and for scientists that need citizen-produced reports to be reliable and "fit" for addressing scientific objectives. We detail some recommendations subsequently in the *Practical implications* section.

## Contribution to the literature on biases in ecological citizen science

An important contribution of this study is in exposing the categories of factors that shape observers' decisions on where, when, and what to observe, as well as which observations to report, namely, *recordability*, *community value*, *personal preferences* and *convenience*. Few prior studies have alluded to these factors [12, 76–78, 81, 90]. In particular, our findings suggest that—at least in the context of CS projects that require a species' photograph to be recorded and shared using smartphone or web upload–*recordability* is the most salient factor that influences observers' choices (in our study, 36% of utterances). The literature makes a distinction between expertise-based CS projects (where trained and certified volunteers monitor a species within a region employing professional monitoring protocols and equipment) and evidence-based projects (such as iNaturalist) [11]. The quality assurance processes in expertise-based projects rely primarily on the expert's (or trained volunteer's) ability to identify focal species and follow monitoring protocols), such that there is no requirement for providing evidence in the form of time-stamped geo-tagged photo. Without the requirement for a photo, *recordability* is not an issue, the hence prior studies of observer behavior in expertise-based CS were concerned with *detectability* (aka "observability") [12, 45, 111–113]. We note that only few prior works explicitly discussed the notion of *recordability* [11, 76, 78].

Our findings also highlight the importance of *community value* (e.g., contributing to the project's goals, species' rarity; 29% of utterances in our study) and *personal preferences* (e.g. favoring particular species, learning; 22% of utterances in our study) in shaping observers' reporting decisions. Finally, our results show that *convenience*-related considerations also affect observers' reports (in our study, 13% of utterances).

We noted substantial differences in observers' decision whether to record an observation (i.e., *recordability*), in the extent to which species rarity influenced their observation patterns (i.e., *community value*), in their affinity for certain species (i.e., *personal preferences*), and in the way various constraints influenced their decisions on where, when, and what to observe, as well as which observations they reported (i.e., *convenience*).

In addition to the four categories identified, our study of observers' decision-making process revealed that their views regarding the monitoring protocol were varied. Some observers reported on a narrow list of species, primarily based on attachment to the species; others were primarily interested in a broad category of species, for instance insects or plants, and within that category attempted to record a large variety of subspecies, and a smaller group of particularly active observers attempted to record every species. Recognizing the risk of bias and the benefits of a more structured monitoring protocol, this small group opted to monitor in a semisystematic manner. This indicates that the majority of participants preferred the flexibility and autonomy enabled by the unsystematic protocol.

In this context, it is worth noting that some shared attitudes and preferences, such as the value that observers place on the communal goal of producing a valuable, high-quality archive of observations, are apt to *reduce*, rather than heighten, temporal, spatial, and taxonomic biases. For example, community-members' awareness of the importance of recording both diurnal and nocturnal animals may lead them to conduct more nighttime observations. Similarly, understanding the importance of collectively developing an accurate "map" of nature may encourage observers to report also on common species. Interestingly, without being asked about bias directly, participants addressed this issue as one aspect of *community value*, reflected specifically in their desire to learn, and to produce a reliable archive of observations.

Our findings further suggest that the effects that participant preferences and communalities have on temporal, spatial, and taxonomic bias differ: First, observers' biases are *not* likely to pose a serious threat to ecologists' ability to identify ***temporal*** trends in the data (e.g., changes of population sizes over time). Given that ecologists often focus on a particular species and study seasonal or yearly trends, communalities of a more granular character (e.g., a preference for weekend vs. weekday observations) are not likely to affect the conclusions about seasonal/ yearly trends. Second, when considering the effect of participant trends and commonalities on spatial bias in CS data, the issue of a localized group of observers, their access to certain terrains or enclosed areas, and their shared ecological concerns regarding a specific ecological threat, are all factors that are likely to result in **spatial bias**. However, in this regard we wish to point out that the systematic monitoring protocols employed by scientists and nature conservation agencies are also affected by—and subject to—accessibility constraints. Their solution is to collect data from a few selected locations (e.g., a transect survey), the trends of which are then extrapolated to surrounding areas. We note here that similar extrapolation methods could be applied when working with CS data. Hence, it may be surmised that the risk of spatial biases in unstructured CS data is *not* particularly high, as compared to the similar risks involved when using systematic monitoring protocols. Finally, on the question of CS-collected data leading to taxonomic or species-related bias, our results demonstrated numerous trends that affect participants' behaviors, which together increase the likelihood of **taxonomic bias** in the accumulated database. Here we mention yet another behavioral trend that is likely to increase taxonomic bias (but was not presented in the results section because it was expressed by only one participant), namely, *perceived danger*. Animals perceived as posing a potential danger to humans are difficult to record. Assuming that people share these fears, the aggregate database of reports is likely to under-represent dangerous animals. Although in the particular context of our study, the animals observed only rarely pose a threat to humans, the fact that

perceived danger may be a salient factor in other CS projects further increases the likelihood that CS-collected data will contain taxonomic bias.

Another novel insight and a contribution of our study is in revealing the instability in observers' attitudes and preferences. Our questionnaire and interviews revealed that temporal shifts in observers' choices are common. Such shifts may be associated with a key event (e.g., moving one's residence or purchasing new photography equipment), learning and developing new interests (an expert in reptiles gradually takes interest in insects), or may reflect change in habit (e.g., change in one's availability during the week, ceasing to monitor regions that are open to the public only during weekends). Such shifts may have key implications for our understanding of observer-based biases and for the distortion they create in the community-generated database of observations.

Together, our findings inform the literature on citizen scientists' perceptions, attitudes, and behavior [11, 12, 67, 114–133]. Knowing what the biases are and understanding their salience is important for the design of interventions [79] and statistical methods [84–86, 134] that attempt to alleviate observer-based biases in CS ecological monitoring projects.

## Contribution to the literature on biases in collective intelligence systems

For the most part, prior works on biases in ecological CS have been restricted to that particular context, thus missing an opportunity to inform, as well as to be informed by, relevant literature in the related field of collective intelligence. A contribution of this study is in placing the discussion of biases in ecological CS within the broader discourse on biases in collective intelligence systems. When considering the reliability of data collected by citizens on a CS platform, and especially a project that does not enforce a strict protocol, we note that such concerns are not limited to CS ecological monitoring, but rather are relevant to other collective intelligence systems.

The literature on biases in user-generated content and social networks has been mostly concerned with large-scale collective intelligence platforms, such as Wikipedia [20, 26–28, 135–141]. In those settings, people's preferences, attitudes, worldview, and expertise determine what content they choose to contribute [23, 137] (referred to as 'motivational biases' [138, 139]). However, because of the group's size, diversity, and the range of members' independent opinions [142, 143], their preferences cancel each other out and thus, bias in the aggregate outcome is attenuated [17–19].

By contrast, the global/local (or *glocal*; [144]) organization of collective intelligence, such as the nodes on the iNaturalist platform, challenges the assumptions of group diversity and independence, and calls into question the platform's ability to successfully distill "wisdom from the crowd", or in other words, valuable, truthful insight from the information that is gathered from local monitoring projects. Clearly, local configurations are important for encouraging members' contribution and engagement, as well as for organizing and coordinating activity [145, 146]. Nonetheless, as our findings indicate, the content generated by such local communities is particularly prone to biases [147], because social networks and geographical collocation help foster common values, beliefs, and concerns, which spread and take hold in groups and organizations [9, 148–150]. For example, the identification of the subcategories of *importance to the community* and *importance to the archive of observations* under the category of *community value* suggest that members of a rural community may ascribe particular importance to the observation of a species that preys on their livestock or harms their crops, resulting in over-representation of that predator in the collective database. In another example, our participants ascribed importance to the deployment of wind turbines, which was endangering the birds in the area. In a similar manner, particular area threatened by future industrial

development may organize to track and record wildlife in that particular location (e.g., bio-blitz), resulting in over-representation of the region and its wildlife.

Furthermore, norms and social pressures may be heightened for people who live in the same small local community, causing people to behave in a similar manner. This is especially relevant to the subcategories of *personal preferences* (i.e., for a particular species, region, or time), as neighbors may organize joint observations at a particular time and/or area, or share their affiliation for a particular species, thus yielding an over-representation of these species in the archive. Hence, despite some diversity in members' preferences and constraints (as discussed above), in the context of a local community, members are likely to share common preferences and constraints, which—when aggregated—yield biases, especially taxonomic biases, in the database of observation.

Another concern that is pertinent when considering biases in both CS and collective intelligence projects is participants' uneven activity patterns. Prior studies have demonstrated that activity distributions within cyber CS projects—and more broadly, in online communities (e.g., open-source software development, Wikipedia)—approximate a power law distribution, whereby the vast majority of peripheral participants contribute only few observations and few highly-active core community members are responsible for a large portion of observations [99, 151–153]. Presumably, this may further increase the likelihood of bias. When the few highly-active participants, especially in a local CS project, share the same preferences, the aggregate database may suffer from biases, despite diversity in the attitudes and preferences of other, less active community members. To sum, we make a fundamental distinction between local CS communities and global environmental monitoring platforms, arguing that aggregate database of observations that are generated by local communities of observers are more likely to be biased.

## Practical implications

Findings from our study have important implications for the practice of CS, beyond the contributions to the scholarly literature that were discussed above. We offer some directions for enhancing the quality of CS data, to be used as a standalone source of biodiversity data, or alternatively as a data source that complements data generated through the use of systematic monitoring protocols [154].

In particular, we point to two key practical implications: the first pertains to the governance of the project and to procedures intended to reduce biases, whereas the second concerns the possibility of statistically adjusting for biases in the citizen-reported data. Is there a way for project leaders and custodians to guide the monitoring process, such that the aggregated data is less biased? Possibly, volunteers could be instructed to follow systematic monitoring protocols, but this would require special training, may have detrimental effects on volunteers' motivation and commitment, and essentially go against the fundamental tenets of opportunistic CS projects such as iNaturalist. Nonetheless, they may be some ways to softly "nudge" volunteer observers, in an attempt to influence their reporting decisions and reduce biases [79], for example, by recommending that they monitor less visited areas. Biases could possibly be moderated by explicitly asking observers to vary their observations in terms of location, time, and species, recommending that observations be performed across the entire region, across seasons, and times of day, and for all species. In addition, project leaders could organize observation events (e.g. bio-blitzes) to target less recorded species, regions, and times (e.g., nighty events).

An alternative strategy to manipulating the community as a whole, which leverages the personal differences that were identified in this study, is to try and control for the composition of

the observer community, for example, by intentionally inviting volunteers that specialize in various species (e.g., some with a special interest in birds, others with an interest in reptiles, etc.), such that in aggregate, all species are covered. Another way of reducing biases is curbing constraints that are associated with the reporting tools, e.g., solving technical problems with the mobile app or encouraging volunteers to use professional photography equipment. We recommend that providing such directions to volunteer observers should be practiced with extra caution, as posing restrictions on the monitoring process could have detrimental effects on their motivation and engagement [31, 67, 129]. Hence, in light of the personal differences in observers' motivation, preferences, and behaviors, it may be most useful to allow for multiple forms of engagement [145]. In other words, it may be useful for the level of guidance to be personalized (e.g., some may prefer more guidance and structure regarding what to record, whereas others may prefer the autonomy to follow their interests).

A second implication involves the attempts to develop statistical methods that would (partially) correct for biases in biodiversity databases produced using opportunistic monitoring procedures. Researchers in the field are concerned that datasets gathered through citizen-science methods often do not accurately represent species' distribution over space and time, and thus may induce errors in models attempting to predict species distribution or abundance patterns [11, 12, 74, 90]. In particular, the likelihood of recording a species within a region is a function of sampling bias, imperfect detection [85], and observers' decisions regarding what to report [12]. For example, Bird et al. [155] demonstrated that not accounting for detection probability resulted in a dramatic underestimate of species abundance and occurrence. In recognition of these issues, scholars have called for statistically controlling for observer-based biases [76, 155–157]. Developing validated methods for correcting sampling bias for citizen-generated data is an active area of research in the species distribution modeling field (e.g. [158]).

In light of our findings regarding the variance in observers' preferences the development of bias-correction methods which account for individual-level biases could be beneficial. However, simple approaches, e.g., controlling for the effects associated with observers' skills and location/time preferences through standardization, may not suffice to eliminate the heterogeneity, as there are other variables that influence species detectability and *recordability*, for example, the effort (or time) spent in each monitoring excursion [159]. Another useful approach may be to cluster observers based on their prototypical psychological and behavioral patterns, along the lines of the approach suggested in other studies [78, 109], and to adjust the bias-correction method per clusters of observers. Over the past decade, there have been preliminary attempts to develop statistical methods for correcting biases in data gathered through opportunistic monitoring [84, 85, 160]. Although these approaches provide a sound starting point to tackle observer-based biases, none of these methods has considered the factors that affect the likelihood of reporting an observation once it has been detected. We propose that complementing these types of statistical methods with data collected regarding volunteers' preferences and attitudes (e.g., using a questionnaire or advanced empirical methods, such as virtual reality simulations) could provide a potential remedy. Just as semistructured monitoring projects, such as eBird, gather metadata about the observation process (e.g., start and end times), collecting information about preferences and attitudes can be utilized to generate a biodiversity archive that serves scientific purposes [81, 161]. Although this would entail extra effort and would require engaging with the community of volunteer observers, we believe that the potential value of such a hybrid approach—i.e., making the vast amounts of citizen-science biodiversity data suitable for scientific research and policy making—outweighs the disadvantages.

## Limitations and suggestions for future research directions

Conclusions drawn from this study should be considered in light of several limitations. First, we investigated one particular case of a nature-monitoring project (Tatzpiteva) on a specific CS platform (iNaturalist). Granted, iNaturalist is probably the largest platform of its kind and Tatzpiteva is a very large project on that platform (the largest in Israel, representing roughly 50% of the national records on the iNaturalist platform); nevertheless, some of the distinct features of our setting may have influenced our findings. For instance, design choices of the iNaturalist platform (specifically: the requirement to upload photos, which constrains observer behavior, as discussed above), as well properties of the project and of the community: Tatzpiteva is general in its purpose, recording all species within a region, and the observers form a tightly-knit local community, rooted within the geographical, societal, cultural context of Israel with its own unique characteristics. When attempting to generalize findings to different citizen science platform, one should consider the project's socio-technical setting, including platform design and the character of the community. Although the factors underlying observers' reporting decisions that were identified in this study are likely to be found in other settings, the relative salience of these factors may differ between CS platforms, projects, and cultural settings. For example, observers' tendency to record observations close to their home is particularly prominent for citizen scientists that reside in rural areas where nature is just outside the door, but is likely to be less salient in urban projects. Hence, we call for future research to investigate observer-based biases in alternative settings.

Second, although the multimethod case-study approach that we employed provided a rounded and comprehensive view of CS practices, the number of participants was not large; it may possible to expand the scope and dive somewhat deeper using a particular method (e.g., including more interviews). Furthermore, although we triangulated our data collection using more than a single method, much of our findings are based on observers' self-reports, and these may not fully reflect observers' actual decision-making processes [162]. Indeed, we found that participants were not entirely consistent in their responses to the various questions. For instance, some participants stated that they report everything that they detect, yet also indicated that a consideration for recording their observations is a personal interest in the species (e.g., [pr4]). There is a need for future research that would explore the linkage between observers' decision-making processes, their actual reporting behavior, and the consequential biases in the aggregate data. For example, future research could conduct a large-scale empirical study to statistically analyze the extent to which various observer considerations predict their reporting patterns, attempting to assign weights to these various biasing factors [110]. An additional interesting avenue for future research is to investigate the motivational processes underlying observers' considerations. Prior research on the motivation for participation in CS projects [67, 163] has employed generic frameworks such as Self Determination Theory [164] or the model for collective action [165]. We suggest that future research move beyond these generic conceptualizations to study the specific motivational factors that are directly linked to observer-based biases. A deeper understanding of the motivational dynamics underlying observers' behavior could yield insights that may be relevant for mitigating the biases.

Additionally, our study investigated observer-based biases within a local community of nature observers, studying *the biases of those who have selected to participate*. Additional biases may stem from the socioeconomic factors that shape the demographics of citizen scientists, i.e., participation bias [12]. For example, people's demographic background greatly influences the way in which they navigate space [166, 167]. To empirically study participation bias, future research is encouraged to expand the analysis of observers' preferences and constraints to other cultures and geographies. Finally, notwithstanding the value of the "use-agnostic"

approach [63, 168] that we have adopted in the current study, we encourage future research to dig deeper into the possible uses of citizen science data and consider the extent to which observer-based biases impact specific ecological research questions, such as species' richness, temporal distribution and population sizes.

We also call for future research that would delve deeper into biases that stem from the project's design. Namely, data quality in CS projects is influenced not only by observer-based biases; importantly, project sponsors goals and design choices can also have significant effects on data collection, and eventually, on the quality of the data. Specifically, a "fitness for use" approach, which is often adopted means that project sponsors might design protocols to meet specific research goals, potentially neglecting broader biodiversity aspects. For example, a project focused on bird species may not gather adequate data on insects or plants [169], whether because of focus of guidance and instructions, or due to prioritizing resource allocation towards certain goals. This may seem like a non-issue at first glance. After all, project is designed to achieve certain goals, it seems only natural that it will be designed accordingly. However, as Follett and Strezov [170] have pointed out, there is a growing number of studies which rely on the re-use of collected datasets from past citizen science research projects. Uses might not necessarily be fully known at the time a project is designed and launched, and may change over time (consider, e.g., Pharr, Cooper [62], where data from a CS project is combined with US government light and noise data to answer questions not considered when the original CS project was designed). In that context, a "use agnostic" perspective [63, 64] can be useful for considering issues of data quality.

Similarly, another source of potential bias is bias induced by the platform design. In particular, iNaturalist employs specific work processes that stem from its evidence-based opportunistic monitoring approach. One of the key factors that emerged in our data as affecting observers' monitoring decisions is *recordability*–the ability to take a picture or record a video. While photographic evidence is useful for enabling verification, and is obviously helpful in species identification, mandating photographic evidence also introduces systemic bias, driven by platform design rather than by observers' preferences. In this case, a bias against species that are smaller, nocturnal, and/or more agile and difficult to capture in a photo. Hence, we encourage future research to investigate the way in which project and platform design choices shape observers' decisions, and consequently biases in the co-created archive of reported observations.

## Conclusion

As the world's ecosystems are undergoing rapid and significant changes, characterized by a continuous decline in biodiversity and in the abundance of insects, birds, and mammals [171, 172], scientists must be able to detect changes and identify warning signs much quicker, in order to develop and aim productive timely conservation activities. However, several factors limit the ability of traditional scientific monitoring methods to detect ecological changes, as they rely on systematic protocols and professionally trained observers, and are costly and difficult to scale [11, 12, 110]. As a consequence, long-term and wide-scale monitoring initiatives are often limited to very few sampling sites within limited regions and to particular times; however, the attempt to generalize from these limited findings to different places and times is problematic. Furthermore, given the budget constraints and scientists' focus on particular species, most of a region's species are not monitored systematically, limiting ecologists' ability to consider interspecies interactions and thus making it difficult to assess long-term trends in the ecological system.

Citizen science, and specifically unstructured CS, has the potential to become an important approach for biodiversity monitoring, which will overcome the limitations of traditional monitoring methods. Prior research has described observer-based biases in CS data [11, 12, 173–175], but less attention has been given to how these biases arise as a result of both social and ecological variables. A major impediment for employing unstructured CS data is that it reflects both the ecological reality at a given time and place and observers' preferences and choices [12]. Without decoupling the two, it would be impossible to determine the state of nature [11].

We studied a local community of nature observers, which operates a project (*Tatzpiteva*, in Israel) using a global collective intelligence platform, iNaturalist. Community members monitor, record, and share geo-and time-tagged images of plants and animals. This project affords observers very high levels of autonomy, allowing them to report on any species they choose, at any place or time, providing limited guidance and direction. Hence, such a setting was likely to reveal a broad range of biases. We collected data through observations, questionnaires, interviews, and archival textual material. Our findings identified four key factors that influence an observer's decisions about what, when, and where to observe, as well as which specimens to report once detected: *recordability*, *community value*, *personal preferences*, and *convenience*. Examining these factors, we demonstrated that local nature observers within a community share common considerations when determining their observations and reporting choices. Thus, we add to the literature on CS, and specifically to our understanding of data-quality issues in ecological monitoring by citizen scientists [11, 12, 34, 36, 40, 41, 68–70]. We make a broader contribution to the scholarly discourse on biases in collective intelligence systems [17, 18, 176–180] by showing that, at least in the context of our study, individuals' reports tend to demonstrate a trend in a particular direction, such that some biases are not cancelled out but rather are amplified, naturally leading to inaccuracies in the collective intelligence system.

Our findings regarding the potential for taxonomic biases in the aggregate database of observations call into question the use of unstructured CS data for determining species' spatiotemporal distribution (e.g., estimating populations sizes). According to Arazy and Malkinson [11], the key for decoupling the ecological process from observers' decision-making process is to elucidate the probability for a specific observer to report on a particular species once encountered. Hence, such a framework is both person- and species-specific. Findings from our study lend support to this approach, as they highlight the intricate array of personal considerations underlying an observer's decision to report a particular species. For such an approach to become practical, future research in two primary directions is warranted. First, it is necessary to develop methods for estimating the probability of an observer-species reporting preference. This may be possible using traditional behavioral research approaches, e.g., a questionnaire [11], or by employing more advanced methods, such as nature immersed Virtual Reality simulations [181, 182]. Either newly developed statistical measures (or proxies) or existing comparative statistical analysis could be used to determine the likelihood of reporting once a species is observed. Another potential avenue for future research is to develop novel statistical methods that would take the abovementioned probability as an input and produce estimates for the state of nature, specifically regarding species' spatiotemporal distribution.

To conclude, CS has great societal benefits in linking people to nature, resulting in a strengthened sense of community, belonging, and caring for the local environment [183–185], as well as greater personal agency [186], environmental advocacy, and activism [187–189]. Beyond the societal benefits, we believe that it is possible to utilize the vast amounts of biodiversity data gathered using online platforms such as iNaturalist for assessing the state of nature. We hope that this study will encourage future research on observers' psychology and behavior, facilitating the development of statistical methods that correct for observer-based biases in unstructured CS data, so as to facilitate scientists' efforts to track trends in the world's

biodiversity. Enhancing our ability to detect trends in species populations can help us to promptly design, tailor, and execute informed interventions that are much needed to protect and sustain the environment.

## Supporting information

**S1 Appendix. Questionnaire.**
(DOCX)

**S2 Appendix. Interview protocol.**
(DOCX)

**S3 Appendix. Code book.**
(DOCX)

**S1 Data.**
(XLSX)

## Acknowledgments

We thank the anonymous reviewers for insightful comments that helped us improve this paper. We thank members of the Tatzpiteva community, and in particular the community leader, Ariel Shamir, for their efforts in recording biodiversity and working to conserve nature, as well as for their contribution to this study. We commemorate Arie Ohad, one of the founders of the community and an active member, who has passed away recently.

## Author Contributions

**Conceptualization:** Ofer Arazy, Dan Malkinson, Yiftach Nagar.

**Data curation:** Keren Kaplan-Mintz.

**Formal analysis:** Ofer Arazy, Keren Kaplan-Mintz, Yiftach Nagar.

**Funding acquisition:** Ofer Arazy, Dan Malkinson.

**Investigation:** Ofer Arazy, Keren Kaplan-Mintz.

**Methodology:** Ofer Arazy, Keren Kaplan-Mintz.

**Project administration:** Ofer Arazy.

**Supervision:** Ofer Arazy.

**Writing – original draft:** Ofer Arazy, Yiftach Nagar.

**Writing – review & editing:** Ofer Arazy, Keren Kaplan-Mintz, Dan Malkinson, Yiftach Nagar.

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
