## [Decision Letter · Decision Letter 0]

10 Mar 2024

PONE-D-23-35664A local community on a global collective intelligence platform: a case study of individual preferences and collective bias in ecological citizen sciencePLOS ONE

Dear Dr. Arazy,

Thank you for submitting your manuscript to PLOS ONE. After careful consideration, we feel that it has merit but does not fully meet PLOS ONE’s publication criteria as it currently stands. Therefore, we invite you to submit a revised version of the manuscript that addresses the points raised during the review process.

We look forward to receiving your revised manuscript.

Kind regards,

Hong Qin

Academic Editor

PLOS ONE

Journal Requirements:

4. Thank you for stating the following financial disclosure: "This research was supported in part by the University of Haifa’s Data Science Research Center."  

6. Please amend your list of authors on the manuscript to ensure that each author is linked to an affiliation. Authors’ affiliations should reflect the institution where the work was done (if authors moved subsequently, you can also list the new affiliation stating “current affiliation:….” as necessary).

Reviewers' comments:

Reviewer's Responses to Questions

**Comments to the Author**

1. Is the manuscript technically sound, and do the data support the conclusions?

Reviewer #1: Partly

Reviewer #2: Partly

2. Has the statistical analysis been performed appropriately and rigorously? 

Reviewer #1: I Don't Know

Reviewer #2: N/A

3. Have the authors made all data underlying the findings in their manuscript fully available?

Reviewer #1: No

Reviewer #2: No

4. Is the manuscript presented in an intelligible fashion and written in standard English?

Reviewer #1: Yes

Reviewer #2: Yes

5. Review Comments to the Author

Reviewer #1: Dear colleagues,

thank you very much for the possibiliy of reviewing this manuscript about potential biases induced by individual and personal attitudes and preferences in an ecological citizen science project. I enjoyed reading the paper and I find the subject very topical for many aspects, from the mere evaluation of citizen science approaches to the indirect relevance of directly involving people in monitoring environmental changes.

In general I find the paper well written and the supporting literature exhaustively mentioned. However, I also find the text flow sometimes unnecessarily redundant (similar considerations in the introduction and in the discusssion e.g.) and to some extent confusing, as far as the captions are concerned: for instance at page 4 (apparently the end of the introduction) where there is a hint to the results ("Our results reveal four primary factors that ...") without having even mentioned the research questions and/or the methodological approach (at this point there is a caption called 'Background', which should be mentioned earlier, maybe). Please check agian the sequence and contents of the sinlge captions. This will help the reader to follow the study.

My major concerns regard the method: the paper reads that "roughly 40,000 observations were reported on Tatzpiteva by 400 observers, making up roughly half of all the iNaturalist observations in Israel. Most of Tatzpiteva’s observations were

contributed by the community’s core members, where most members are peripheral and contribute only occasionally."

Would it be possible to be more precise about the so called core members (how many are they and how many observations did they deliver?) as well as about the so called peripheral members? Are the core members the 38 persons that were interviewed?

It would be appreciated if the entire questionnaire would be made availale as a supplementary material (e.g.) instead of listing the questions in the text flow. Regarding the questionnaire, it is not fully clear to me, why only one species was chosen as higly likely to be observed whereas three species were listed as higly unlikely -why not having two likely and two unlikely species?

Table 1 show a very broaden distribution of the age and the contributions (number of reported observations) of the 27 participants: I am wondering (i) if it would be possible to add a hint to the standard error (or standard deviation) for both these measurements and most important (ii) how representative the sample is (27 out of 400 observers in sum, this is less than 10%; a maximum of approx. 3,500 reproted observation out of a sum of 40,000, the same here). I would appreciate reading some considerations about this in the discussion.

I am not a social scientist and I am not confident with citing answers from questionnaires in a paper, so while begging for understanding for my ignorance, I wonder (again) to which extent citing single statements of a low number of participants is representative for the study topic.

In the discussion I would appreciate some considerations for possible biases in studies on similar topics conducted by scientists instead of interested volunteers, for instance, and how combining the two approaches could maybe generate a win-win situation for everybody.

Minor comments:

. please check the references once more, some of them did not get a number and are included in the text flow (see for instance Hochmair et al., 2020 at page 2 and Arazy & Malkinson, 2021 (page 27) in the discussion.

. please carefully check the language again - sometomes I had the feeling that the sentences were incomplete (but I am not a native speaker, so I might be biased). E.g. page 2: "In particular, and the local organization of these communities could possibly hinder their ability to produce unbiased outcomes."

Reviewer #2: I read this manuscript with great interest. The authors report qualitatively on a multi-year citizen science project to collect data about the flora and fauna of Israel. The key focus of this manuscript is on biases in data arising from characteristics of contributors in deciding which observations to report. Overall, this manuscript provides useful qualitative insights into "what makes citizen scientists "tick" in the context of a specific project. The four characteristics of recordability, collective considerations, personal preferences, and convenience are intuitive and useful. The detailed comments from contributors help contextualize the findings.

I do have several concerns and suggestions for improvement:

1. There seems to be an underlying assumption that collectively, contributions will "cancel out" biases of individual contributors. However, I do not see the validity of using this as a starting point. Geographic biases (e.g., concentrations near towns, roads, etc.) will not be overcome necessarily by having more contributors, nor will temporal biases, or biases toward reporting rare species or species that are hard to locate (e.g., nocturnal). These limitations seem built in to many CS projects. It would be useful, though, if the paper proposed strategies to mitigate such biases (e.g, design decisions, instructions to contributors).

2. The project seems to require participants to upload photos, with a reason for not contributing being things like animals appearing fleetingly so that photos could not be captured. Why not allow contributors to report observations using text? A fast moving mammal or bird might not afford a photo oppotunity, but contributors certainly could make such reports (thereby mitigating bias!).

3. The paper seems to be hastily written. There are numerous typos (no space to list them here), and the paper uses strange word choices at time (notably, "slant", which seem to be a very informal work choice for a scientific paper. Why not stick to bias?

6. PLOS authors have the option to publish the peer review history of their article (what does this mean?). If published, this will include your full peer review and any attached files.

Reviewer #1: No

Reviewer #2: No

---

## [Author Response · Author response to Decision Letter 0]

22 Apr 2024

Please see detailed responses in the attached rebuttal letter

---

## [Decision Letter · Decision Letter 1]

30 May 2024

PONE-D-23-35664R1A local community on a global collective intelligence platform: a case study of individual preferences and collective bias in ecological citizen sciencePLOS ONE

Dear Dr. Arazy,

Thank you for submitting your manuscript to PLOS ONE. After careful consideration, we feel that it has merit but does not fully meet PLOS ONE’s publication criteria as it currently stands. Therefore, we invite you to submit a revised version of the manuscript that addresses the points raised during the review process.

We look forward to receiving your revised manuscript.

Kind regards,

Hong Qin

Academic Editor

PLOS ONE

Journal Requirements:

Reviewers' comments:

Reviewer's Responses to Questions

**Comments to the Author**

1. If the authors have adequately addressed your comments raised in a previous round of review and you feel that this manuscript is now acceptable for publication, you may indicate that here to bypass the “Comments to the Author” section, enter your conflict of interest statement in the “Confidential to Editor” section, and submit your "Accept" recommendation.

Reviewer #1: All comments have been addressed

Reviewer #2: (No Response)

2. Is the manuscript technically sound, and do the data support the conclusions?

Reviewer #1: Yes

Reviewer #2: Yes

3. Has the statistical analysis been performed appropriately and rigorously? 

Reviewer #1: Yes

Reviewer #2: N/A

4. Have the authors made all data underlying the findings in their manuscript fully available?

Reviewer #1: Yes

Reviewer #2: Yes

5. Is the manuscript presented in an intelligible fashion and written in standard English?

Reviewer #1: Yes

Reviewer #2: Yes

6. Review Comments to the Author

Reviewer #1: The manuscript has consistently improved since last time and I acknowledge the thoroughful revision done by the authors. There are no further comments from my side and I am looking forward to seeing the work published.

Reviewer #2: I thank the authors for their revision and the clear response to the reviews in the previous round. I continue to like the paper and have only three remaining substantive comments, both arising from the response/revisions:

1. Framework for Biases: The topic of bias in citizen-generated data is important. A recent article proposed a framework for understanding "socio-ecological biases" in citizen science data (Carlen et al., 2024). Given the apparent relevance to your work, it might help to relate your perspective to their framework. I don't think they have "scooped" you, but on a first glance, your in-depth qualitative work can provide support for aspects of their framework and perhaps challenge others.

2. Data Quality: As noted on page 4, traditional approaches to data quality focus on fitness-for-use and, to assess this, "we must consider the possible uses." However, it may be that uses are not be fully known at the time a project is designed and launched and/or may change (consider, e.g., Pharr et al., 2023, where data from a CS project is combined with US government light and noise data to answer questions not considered when the original CS project was designed). As you note, "CS data can be valuable for addressing a variety of research questions." In that context, there has been a suggestion that data quality in citizen science (and crowdsourcing more broadly) can sometimes be considered from a "use-agnostic" perspective (Wiersma et al., 2024; Lukyanenko et al., 2014). In that context, it is unclear how contributors' preferences (versus the goals of project sponsors) affect the nature of bias and resulting impacts on data quality. For example, project sponsors likely have specific goals (e.g., understanding the distribution and prevalence of (focal) species), whereas contributors may produce biases along the dimensions identified in your analysis. Some discussion of the inherent tension between viewing data quality from a fitness-for-use perspective and the way in which data can become biased as a results of that focus imposed by project goals (in addition to, and perhaps in different ways from, the biases you discuss in the paper based on behavior and preferences of contributors) would deepen the analysis of the relationship between bias and data quality.

3. Platform-design-induced bias: The only response to my previous comments that was less that satisfying was the one regarding the photo requirement. I realize this is iNaturalist-driven, rather than Tatzpiteva-driven, but it seems clear that such a requirement is a big source of systematic bias. One of your key findings is that "recordability" drives the decision to (not) report, or in the words of one of your respondents "the ability to take a picture." Unlike issues related to personal preferences or convenience, this seems an artifact of underlying decisions of the platform. While a photo might be useful in enabling verification, it seems a mechanism to induce severe bias in certain contexts (e.g., low light, fast-moving animals) by preventing participants from reporting observations they might otherwise be inclined to report. As Lukyanenko et al. (2019) show, it is possible to perform post hoc processing on textual data to get a high level of classification performance on textual data (no photos) that lacks species identification.

In conclusion, this paper stands to make a strong contribution to our understanding of preferences and biases among contributors to citizen science projects. The comments above are intended in the spirit of better refining the message and contextualizing it with respect to related work.

References

Carlen, Elizabeth J, Cesar O Estien, Tal Caspi, Deja Perkins, Benjamin R Goldstein, Samantha ES Kreling, Yasmine Hentati, Tyus D Williams, Lauren A Stanton, Simone Des Roches, Rebecca F Johnson, Alison N Young, Caren B Cooper, Christopher J Schell (2024). A framework for contextualizing social-ecological biases in contributory science. People and Nature, 6(2), 377-390.

Lukyanenko, Roman, Jeffrey Parsons, Yolanda Wiersma, Mahed Madadah (2019). Expecting the unexpected: effects of data collection design choices on the quality of crowdsourced user-generated content. MIS Quarterly, 43(2), 623-647.

Pharr, Lauren D, Caren B Cooper, Brian Evans, Christopher E Moorman, Margaret A Voss, Jelena Vukomanovic, Peter P Marra (2023). Using citizen science data to investigate annual survival rates of resident birds in relation to noise and light pollution. Urban Ecosystems 26(6), 1629-1637.

Wiersma, Yolanda F, Tom Clenche, Mardon Erbland, Gisela Wachinger, Roman Lukyanenko, Jeffrey Parsons (2023). Advantages and Drawbacks of Open-Ended, Use-Agnostic Citizen Science Data Collection: A Case Study. Citizen Science: Theory and Practice, 9(1), 5pp.

7. PLOS authors have the option to publish the peer review history of their article (what does this mean?). If published, this will include your full peer review and any attached files.

Reviewer #1: No

Reviewer #2: No

---

## [Author Response · Author response to Decision Letter 1]

15 Jul 2024

Please see attached Rebuttal letter

---

## [Decision Letter · Decision Letter 2]

26 Jul 2024

A local community on a global collective intelligence platform: a case study of individual preferences and collective bias in ecological citizen science

PONE-D-23-35664R2

Dear Dr. Arazy,

We’re pleased to inform you that your manuscript has been judged scientifically suitable for publication and will be formally accepted for publication once it meets all outstanding technical requirements.

Kind regards,

Hong Qin

Academic Editor

PLOS ONE

Reviewers' comments:

Reviewer's Responses to Questions

**Comments to the Author**

1. If the authors have adequately addressed your comments raised in a previous round of review and you feel that this manuscript is now acceptable for publication, you may indicate that here to bypass the “Comments to the Author” section, enter your conflict of interest statement in the “Confidential to Editor” section, and submit your "Accept" recommendation.

Reviewer #2: All comments have been addressed

2. Is the manuscript technically sound, and do the data support the conclusions?

Reviewer #2: Yes

3. Has the statistical analysis been performed appropriately and rigorously? 

Reviewer #2: N/A

4. Have the authors made all data underlying the findings in their manuscript fully available?

Reviewer #2: Yes

5. Is the manuscript presented in an intelligible fashion and written in standard English?

Reviewer #2: Yes

6. Review Comments to the Author

Reviewer #2: The authors have addressed the remaining concerns I identified in the previous round. This is a solid paper with the potential to make an important impact in understanding bias in citizen science data.

7. PLOS authors have the option to publish the peer review history of their article (what does this mean?). If published, this will include your full peer review and any attached files.

Reviewer #2: No

---

## [Editor Report · Acceptance letter]

14 Aug 2024

PONE-D-23-35664R2 

PLOS ONE

Dear Dr. Arazy, 

I'm pleased to inform you that your manuscript has been deemed suitable for publication in PLOS ONE. Congratulations! Your manuscript is now being handed over to our production team.

Kind regards, 

on behalf of

Dr. Hong Qin 

Academic Editor

PLOS ONE